# Reference Values of Spatial and Temporal Gait Parameters in a Contemporary Sample of Spanish Preschool Children: A Cross-Sectional Study

**DOI:** 10.3390/children9081150

**Published:** 2022-07-30

**Authors:** Pedro Ángel Latorre-Román, Juan Antonio Párraga-Montilla, Alejandro Robles-Fuentes, Luis Enrique Roche-Seruendo, Manuel Lucena-Zurita, Marcos Muñoz-Jiménez, Daniel Manjón-Pozas, Jesús Salas-Sánchez, Filipe Almeida da Conceição, Pedro José Consuegra González

**Affiliations:** 1Department of Didactic of Music, Plastic and Body Expression, University of Jaén, 23071 Jaen, Spain; platorre@ujaen.es (P.Á.L.-R.); consuegragonzalezpj@gmail.com (P.J.C.G.); 2Ayuntamiento de Santiago Pontones, 23091 Jaen, Spain; saltolacomba@gmail.com; 3Faculty of Health Sciences, Campus Universitario, Universidad San Jorge, 50830 Zaragoza, Spain; leroche@usj.es; 4Department of Didactic of Music, Plastic and Body Expression, University of Sagrada Familia de Úbeda, Úbeda, 23400 Jaen, Spain; mlucena@fundacionsafa.es (M.L.-Z.); dmanjon@fundacionsafa.es (D.M.-P.); 5Ayuntamiento de Jaén, 23071 Jaen, Spain; mmjimene@ujaen.es; 6Universidad Autónoma de Chile, Santiago 7500912, Chile; jsalas@ujaen.es; 74 LABIOMEP-Porto Biomechanics Laboratory, Centre of Research, Education, Innovation and Intervention in Sport, Faculty of Sports, University of Porto, 4200-450 Porto, Portugal; filipe@fade.up.pt

**Keywords:** pediatrics, gait, normative values, growth

## Abstract

The aim of this study was to analyze the influence of age and sex on kinematic gait parameters in preschool children, and derive reference values for this population. A total of 383 preschool children (age 3–5 years; 207 girls, 176 boys) participated in this study. We used the OptoGait system to assess the kinematics of gait at a comfortable and self-determined speed. No significant differences between the sexes were found for the main gait parameters. Among the participants, there was a significant increase in double support, reductions in absolute cadence and the coefficient of variation (CV) of cadence, an increase in absolute step length (SL), and an increase in the walk ratio (WR) from 3 to 5 years of age. However, the normalized SL and normalized WR displayed a significant reduction in both sexes. Partial correlation indicated a significant association of age with SL and normalized SL, and WR. Additionally, WR showed a significant correlation with the CV of cadence. To summarize, no relevant differences in gait performance were found according to sex; however, age was found to influence gait maturation. The normative values established for Spanish preschool children can be used to monitor healthy gait development.

## 1. Introduction

The preschool age is characterized by important changes in the acquisition of locomotor skills and nervous system maturation [1]. Among these, walking is an important skill for preschool children because its impact is multidimensional, affecting cognitive, social, and later motor development [2]. Learning to walk is a long process throughout childhood, although the greatest improvement occurs during the early years [3].

Several previous studies have attempted to characterize gait in children [3,4,5,6] based on kinematic parameters such as velocity, stride and step length (SL), cadence, step time, gait asymmetries, gait variability, or the angle of foot placement. However, the role of maturation in the variation in gait parameters with age is still unclear [7]; today, how and at what age gait maturity arises are debated [4]. During childhood, walking progresses from an independent but very unstable gait at infancy to adult-like gait patterns around 8 years of age that show the maturation of step speed, SL, and stride length; and the temporal variability in step-cycle duration [8]. In terms of gait development, typically developing children show variability in kinematic gait parameters for a certain period of time [9]. In particular, gait patterns during the early years, such as a wide base of support, small steps, and the prevalence of a double support phase, can be considered as functional adaptations to compensate for instability; these parameters continue maturing toward the adult patterns between 3.5 and 7 years of age.

Overall, stride length and cadence are the main kinematic parameters used to analyze gait maturity. Stride and cadence are functions of anthropometric characteristics and sex [10]. At free speed, stride and SL increase and cadence decreases until gait maturity [4]. Sutherland et al. indicated that the relationships between spatial and temporal parameters are fixed by the age of 4 years [11]. Moreover, Grieve and Gear [12] showed that age 5 is a relevant milestone in gait maturation; however, other authors noted that the relationships between spatial and temporal gait parameters, measured by the walk ratio (stride length/cadence), appears be a feature of gait that matures beyond the age of 7 years [13]. In this regard, the temporal structure of gait variations is not totally established in 7-year-old children; however, in older children (11 to 14 years), the stride dynamics approach the values detected in adults [14]. Whereas some of these changes are due to changes in body size, others can be caused by maturation of motor control [11].

The mean values of kinematic parameters reflect the functional characteristics of gait [3]; however, the variability in these factors provides information regarding the control of gait. Nevertheless, gait function and gait control are not always associated [15]. It would therefore be valuable to understand to what extent the maturity of gait control follows the maturity of functional parameters [3]. In this regard, intrinsic gait variability, i.e., fluctuations in the regularity of gait patterns between repetitive cycles, is intrinsic to the sensorimotor system and influenced by aspects such as age and pathology [16]. Therefore, variability in spatiotemporal parameters can be considered an indirect measure of gait stability [17] and is informative when investigating the development of mature gait [18]. Another factor that can influence children’s gait maturity is the walk ratio (WR). Whereas a vast number of combinations of SL and cadence can occur when walking at a particular speed, adult humans habitually walk at an invariant ratio of SL divided by step rate (WR) [19]. It is a speed-independent index of gait control and reveals energy expenditure, balance, step-to-step variability, and attention demand [20]. The identification of such a parameter may contribute to our understanding of the development of the control of gait in children [21].

Changes in gait patterns occur as a child matures, but they may also reflect pathological changes in neurological or musculoskeletal systems [22]. Previous studies have shown that children with developmental disorders of coordination, attention deficit hyperactivity disorder, or autism generate movement patterns with greater variability and complexity than children with typical development [23,24,25]. In addition, the spatial–temporal gait variables are affected in children with disorders; for example, there is a reduction in cadence, gait velocity, and SL, and an increase in step width in children with autism [26]. Additionally, children with Down syndrome have a shorter stride [27] while children with development coordination disorders walk with shorter steps and at a higher frequency than typically developing children [28]. Therefore, the assessment of spatiotemporal sub-components of gait in children offers significant clinical potential to understand both typical and atypical brain development [29].

In the first years, any attempt to categorize gait development should include a standardized baseline of kinematic parameter values with narrow age intervals in order to provide precise normative data [30]. Therefore, one of the problems in understanding gait maturation is the availability of age- and sex-matched reference values for preschool children. Therefore, it is important to build a reference database for comparison of variations over time in a child’s gait parameters, and normative data on gait are necessary for clinical practice—principally in children whose gait pattern changes over time [7]. Most of the studies in the literature have included large groups of children, with ages ranging from 5 to 13 years; however, the role of development in the variation in gait parameters with age is still imprecise [7], especially in the early years, during which it is especially important to characterize an individual’s gait and understand how the gait pattern is acquired during childhood. To date, the few quantitative preschool gait studies described in the literature have employed few participants and used expensive equipment, and have been deficient in information regarding the relationships between gait growth and motor development [3,4,30].

Because gait deviations commonly occur in children, in this regard, examination and identification of gait abnormalities allow appropriate clinical decisions to be made. Therefore, developmental centile charts of gait measures offer a solution to assess an individual child’s evolution. Because collections of normative gait data in healthy children significantly diverge from one country to another [7], we decided to produce a specific reference database of gait parameters for Spanish preschool children. Therefore, the main objective of this study was to examine the influence of age and sex on gait kinematic parameters in preschool children, and define the profile of the change in gait variability with age to develop reference values in this population.

## 2. Materials and Methods

### 2.1. Participants

We conducted a cross-sectional study. A prior sample size calculation was performed using G*Power software [31]. The resulting parameters were selected for analysis of variance (ANOVA): an effect size of f = 0.250, an α level of 0.05, a power level of 0.95, three groups, a critical F = 3.032. The sample size was determined to be at least 252 participants. Finally, a total of 383 typically developing children were included in the current study (age = 4.17 ± 0.76 years; age range = 3–5 years, BMI = 15.69 ± 2.59 kg/m^2^; 207 girls and 176 boys). The sample was selected from a large region of Andalusia (Spain) containing both urban and rural populations. The inclusion criteria for participant selection were: the absence of any neurodevelopmental or neuromotor disability, such as autism or Down syndrome; the presence of any pathological disorder associated with the visual or vestibular systems; and orthopedic or developmental difficulties. Parents signed an informed consent form permitting their children to take part in this research. The study was completed per the norms of the Declaration of Helsinki (2013 version) and was accepted by the Ethics Committee at the University of Jaen (Spain) (DIC.18/3.TES).

### 2.2. Materials and Testing

Body mass (kilograms) was measured using a weighing scale (Seca 899; Hamburg, Germany), and body height (centimeters) was assessed with a stadiometer (Seca 222; Hamburg, Germany). The body mass index (BMI) was estimated by dividing weight (kilograms) by body height^2^ (in meters).

Gait speed (GS) was analyzed with photocells (WITTY, Microgate Srl; Bolzano, Italy; 0.001 s accuracy) that were placed at the beginning and end of a 5 m corridor (Figure 1a). We used an OptoGait system (Microgate Srl; Bolzano, Italy) (Figure 1b) to evaluate the kinematics of gait. OptoGait is an optical data acquisition system composed of a transmitter and a receiver bar. Each 1 m bar contains 96 infrared LEDs (1041 cm resolution) and is located on the transmitter bar, continuously communicating with the LEDs positioned on the receiver bar. The bars measure flight and contact times during execution with an accuracy of 1/1000 s. Regarding the reliability of the OptoGait system, all variables analyzed revealed high concurrent validity, with ICCs ranging between 0.933 and 0.999 [32]. The parameters selected for this study were normalized in relation to body height according to a previous study [33]. Five kinematic parameters were analyzed: cadence, SL, WR, and single and double support. The WR represents the relationship between the SL and the cadence of movement of the legs. It is calculated as the mean SL divided by the cadence (steps per minute). Moreover, the variability in SL and cadence in terms of the coefficient of variation (CV) among participants, given as a percentage SD/mean × 100, was analyzed.

### 2.3. Procedure

The assessment was carried out on a flat, straight, noncarpeted surface, and both ends of the walkway were designated by two cones. The subjects were asked to walk at a comfortable and self-determined speed along a 5 m hallway. All subjects wore their own shoes and were asked to walk as naturally as possible. Additional meters at the start and turns were added to avoid recording acceleration or deceleration during walking. Partially recorded footfalls were omitted from the analyses. Children performed three walks for familiarization at a self-selected speed. Ten consecutive walking trials were recorded with an average step count of 56.12 ± 7.49 steps to provide an accurate representation.

### 2.4. Statistical Analysis

Data were analyzed using SPSS, v.19.0 for Windows (SPSS Inc.; Chicago, IL, USA) and the statistical software package R—a free software environment for statistical computing and graphics that compiles and runs on a wide variety of UNIX platforms, Windows and MacOS (R Core Team, 2016, available at: URL https://www.R-project.org/, accessed on 1 September 2021)—with the GAMLSS package and MedCalc Software (Mariakerke, Belgium). The significance level was set to α < 0.05. The data are shown in descriptive statistics for means, standard deviations, and centiles. Tests for the normal distribution and homogeneity of variances (Kolmogorov–Smirnov and Levene’s test, respectively) were conducted on all data before the analysis. Differences between sex and age groups were analyzed using a two-way ANOVA corrected by the Bonferroni test. A partial correlation analysis was performed between gait parameters (adjusted by age and sex); the magnitude of the correlation among measurement variables was set according to Hopkins et al. [34]. The coefficient of variation (CV, %), given as a percentage SD/mean × 100, was calculated as a measure of kinematic variability. For the SL (centimeters), cadence (steps/second),and its CV, the percentile curves were calculated as a function of age stratified by sex using several methods for developing age-related curves. The Lambda, Mu, and Sigma method (LMS) offers an approach to model data with consideration of μ as the location parameter (median), as well as σ as the scale parameter (coefficient of variation), and the skewness parameter λ as the shape parameter. This method was implemented in the GAMLSS package in R software.

## 3. Results

The kinematic parameters of gait are shown in Table 1 in relation to age and sex. Across the entire participant sample, there were significant differences between sexes in double support and the CV of cadence, with girls displaying lower scores. Taking into account the effect of age, both in boys and girls, there was a significant increase in double support (*p* < 0.05), reductions in the absolute cadence (*p* < 0.05) and the CV of cadence (*p* < 0.05), and increases in absolute SL (*p* < 0.001) and the WR (*p* < 0.001) from 3 to 5 years. However, normalized SL and WR displayed a significant reduction (*p* < 0.001 and *p* < 0.01, respectively) in both sexes.

The partial correlation indicated a significant association of age with SL and normalized SL (r = 0.323, *p* < 0.001 and r = −0.400, *p* < 0.001, respectively) and with WR (r = 0.329, *p* < 0.001). In addition, WR showed a significant association with the CV of cadence (r = −0.303, *p* < 0.001). The centiles of cadence, CV of cadence, SL, and CV of SL by age and sex are shown in Table 2 and Table 3.

## 4. Discussion

The main objectives of the current study were to examine the influence of age and sex on kinematic gait parameters in preschool children, to determine the profile of change in gait variability with age, and to derive reference values in Spanish preschool children. The main finding of this study was that non-significant differences were found between sexes in the main gait parameters analyzed; however, age showed a significant influence on cadence, SL and its variability, and WR. Taking into account that preschool age is a period of rapid developmental changes, in disagreement with [30], we can say that the changes in kinematic parameters happen as severely as expected in young typically developing children. In this regard, Rose-Jacobs [4] indicated that because adults walk with a high consistency of gait parameters across a diversity of speeds, this author also expected children with a more mature gait to exhibit less variability in gait factors across a variety of speeds. Their findings, in accordance with the current study, corroborated this hypothesis.

Consistent with previous studies [5,7,22], we found that absolute SL increases with age, while absolute cadence has been shown to decrease. However, when both parameters were normalized, cadence did not show significant changes from 3 to 5 years, while normalized SL displayed a significant reduction. Moreover, only the CV of cadence showed a significant reduction in both sexes from 3 to 5 years. In this regard, Voss et al. [35] showed that after normalization to height, stride-length variability declined with age and was highest in children ages 5−6 compared with all older age groups. Additionally, absolute cadence reduced with age, with children aged 5–10 years walking more steps per minute than older age groups. In the same way, a previous study noted that variability decreased throughout childhood, more rapidly before the age of seven but continuing beyond this age, indicating that gait control was not yet entirely mature [3]. Likewise, Hausdorff et al. [14] noted that measurements of stride-to-stride variability were significantly larger in both 3- and 4-year-old children compared with 6- and 7-year-old children. However, other authors did not find significant differences in normalized spatiotemporal parameters from 2 to 4 years [30]. Therefore, more research on this matter needs to be undertaken before the association between gait variability and childhood growth is more clearly understood.

Overall, WR, as a measure of gait maturation assessment, has often been overlooked in preschool children. WR is commonly a constant value in normal healthy populations, independent of age, height, sex, and gait speed [19,20,36]. A deviation from this constant might show an anomalous walking pattern [36] and may be an indicator of a cautious gait, poor balance control, or an impaired gait [37]. In the case of impaired motor control, WR may be reduced at any speed [20]. Although a decrease in the WR may be interpreted as a nonspecific adaptive mechanism enabling the neural–mechanical control of walking at the expense of a moderate increase in the metabolic cost, an increase in the WR toward normal limits should be taken as descriptive of an improvement in motor control, whatever the underlying mechanisms [20]. In the current study, WR displayed an increase from 3 to 5 years, both in boys and girls; moreover, WR noted a moderate positive correlation with age and a negative one with the CV of cadence. Therefore, the WR may be an idiosyncratic gait characteristic that continues to mature to 11 years [13]. However, additional work is needed to establish whether WR is a strong and sensitive marker of gait maturity, which can contribute to a better understanding of the development of gait control [21].

This study has several limitations. First, and according to Guffey et al. [30], children walking at a self-selected speed could have had differences in kinematic parameters. Alternatively, walking at a free speed would best reproduce each subject’s typical gait pattern rather than using a predefined speed in the laboratory setting, which would result in a less natural gait. Second, gait at several speeds was not tested, and this can better characterize infant gait. Third, the findings and suggestions of this research should be carefully generalized beyond the group of participants analyzed. Finally, we did not consider designing a longitudinal exploration due to the challenges of following children for years, given that most of them move to other schools when they grow up. Nevertheless, this study has several strengths, because the sample comprised many children from a large region, including rural and nonrural areas. In addition, we included in the study of the gait of these children the analysis of the WR. Therefore, to the best of our knowledge, this is the first study with these characteristics carried out in Spanish preschool children.

From a practical point of view and considering the lack of reference values for assessing the gait parameters of Spanish preschoolers, the extreme percentiles can be used as a ‘warning sign’, indicating that it may be necessary to conduct supplementary tests to identify potential motor delays in this population in relation to healthy gait development.

## 5. Conclusions

To summarize, no relevant differences in gait characteristics were found according to sex; however, age was shown to have an influence on gait maturation. The reference values established for Spanish preschool children in the current study could be used to monitor healthy gait development.

## Figures and Tables

**Figure 1 children-09-01150-f001:**
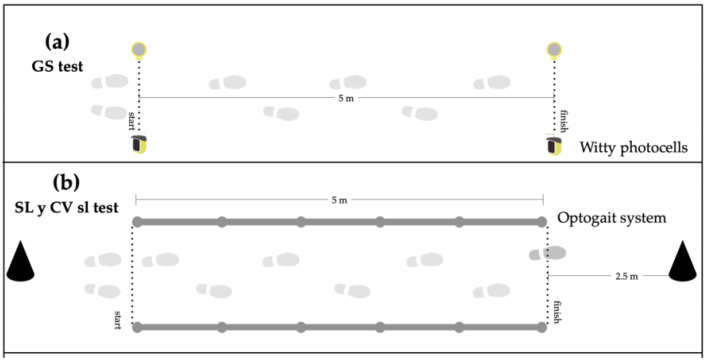
Gait analysis procedure. (**a**) GS: Gait speed, (**b**) SL: step length; CV sl: gait variability of SL.

**Table 1 children-09-01150-t001:** Kinematic parameters of gait in relation to age and sex.

	All Boys	3 Years	4 Years	5 Years	*p*-Value	Post Hoc Analysis	All Girls	3 Years	4 Years	5 Years	*p*-Value	Post-Hoc Analysis
Velocity (m/s)	1.22 (0.18)	1.20 (0.15)	1.20 (0.17)	1.26 (0.19)	0.200		1.24 (1.17)	1.26 (0.17)	1.20 (0.18)	1.28 (0.14)	0.010	4 < 5 *
Velocity normalized	0.37 (0.05)	0.38 (0.05)	0.37 (0.05)	0.37 (0.06)	0.505		0.38 (0.05)	0.40 (0.05)	0.37 (0.05)	0.38 (0.04)	0.008	4 < 5 *
CV Velocity	0.13 (0.06)	0.15 (0.07)	0.13 (0.06)	0.12 (0.06)	0.474		0.12 (0.09)	0.13 (0.06)	0.12 (0.07)	0.13 (0.13)	0.868	
Single support (s)	0.34 (0.04)	0.32 (0.04)	0.34 (0.04)	0.34 (0.04)	0.188		0.34 (0.05)	0.33 (0.05)	0.33 (0.03)	0.35 (0.05)	0.027	4 < 5 *
Double support (s)	0.14 (0.05)	0.13 (0.04)	0.14 (0.04)	0.16 (0.06)	0.018	4 < 5 *	0.13 (0.05) ≠	0.13 (0.05)	0.12 (0.04)	0.15 (0.05)	0.003	4 < 5 **
Cadence (steps/s)	2.44 (0.27)	2.53 (0.29)	2.45 (0.23)	2.37 (0.29)	0.011	3 > 5 *	2.45 (0.27)	2.53 (0.25)	2.50 (0.21)	2.36 (0.21)	<0.001	3 > 5 ** 4 > 5 **
Cadence normalized	0.81 (0.08)	0.81 (0.09)	0.81 (0.07)	0.81 (0.09)	0.944		0.81 (0.08)	0.80 (0.08)	0.82 (0.07)	0.80 (0.10)	0.147	
CV cadence (%)	21.57 (12.76)	25.48 (13.44)	21.78 (13.74)	19.05 (10.75)	0.035	3 > 5 *	18.70 (9.30)	21.77 (13.02)	18.90 (8.30)	16.91 (7.42)	0.026	3 > 5 *
SL (cm)	49.21 (5.39)	47.89 (5.28)	48.43 (5.68)	50.76 (4.83)	<0.001	3 < 4 ** 3 < 5 *** 4 < 5 **	49.83 (5.35)	49.43 (5.23)	48.17 (5.11)	51.66 (5.12)	<0.001	3 < 5 *** 4 < 5 ***
SL normalized	45.17 (5.55)	47.99 (5.81)	44.92 (5.48)	43.76 (4.88)	<0.001	3 > 4 ** 3 > 5 ***	46.24 (5.37)	49.47 (5.17)	45.44 (5.27)	45.34 (4.96)	<0.001	3 > 4 ** 3 > 5 *** 4 > 5 *
CV SL (%)	10.60 (4.80)	12.04 (4.88)	10.54 (4.81)	9.82 (4.63)	0.059		10.65 (7.10)	11.26 (5.42)	10.25 (5.40)	10.73 (9.10)	0.418	
WR	0.34 (0.05)	0.32 (0.05)	0.33 (0.05)	0.36 (0.04)	<0.001	3 < 5 *** 4 < 5 **	0.34 (0.06)	0.33 (0.05)	0.32 (0.04)	0.37 (0.07)	<0.001	3 < 5 *** 4 < 5 ***
WR normalized	0.93 (0.15)	1.00 (0.17)	0.93 (0.15)	0.90 (0.12)	0.001	3 > 4 * 3 > 5 **	0.96 (0.17)	1.03 (0.16)	0.93 (0.14)	0.96 (0.19)	0.008	3 > 4 * 3 > 5 *

≠ significant differences (*p* < 0.05) with boys. * *p* < 0.05, ** *p* < 0.01, *** *p* < 0.001. Values are displayed using the mean and standard deviation. WR: walk ratio. CV: coefficient of variation. SL: step length.

**Table 2 children-09-01150-t002:** Centiles of cadence and CV by age and sex.

	**Cadence (Steps/s)**													
**Age (y)**	**C2**		**C10**		**C25**		**C50**		**C75**		**C90**		**C98**	
	**Boys**	**Girls**	**Boys**	**Girls**	**Boys**	**Girls**	**Boys**	**Girls**	**Boys**	**Girls**	**Boys**	**Girls**	**Boys**	**Girls**
**3**	2.06	2.01	2.21	2.21	2.35	2.36	2.52	2.53	2.71	2.70	2.92	2.85	3.23	3.05
**3.5**	2.00	2.06	2.17	2.24	2.31	2.37	2.48	2.52	2.65	2.67	2.81	2.81	3.02	2.98
**4**	1.94	2.05	2.13	2.22	2.27	2.35	2.44	2.50	2.60	2.64	2.74	2.78	2.92	2.94
**4.5**	1.89	1.94	2.08	2.13	2.23	2.28	2.40	2.45	2.57	2.61	2.73	2.76	2.93	2.95
**5**	1.86	1.76	2.03	1.99	2.18	2.17	2.36	2.38	2.55	2.58	2.75	2.76	3.03	2.99
	**CV of Cadence**													
**Age (y)**	**C2**		**C10**		**C25**		**C50**		**C75**		**C90**		**C98**	
	**Boys**	**Girls**	**Boys**	**Girls**	**Boys**	**Girls**	**Boys**	**Girls**	**Boys**	**Girls**	**Boys**	**Girls**	**Boys**	**Girls**
**3**	0.08	0.08	0.11	0.11	0.15	0.14	0.21	0.18	0.30	0.26	0.42	0.37	0.67	0.64
**3.5**	0.08	0.07	0.11	0.10	0.14	0.13	0.20	0.18	0.28	0.24	0.39	0.32	0.64	0.46
**4**	0.07	0.07	0.10	0.09	0.13	0.12	0.18	0.17	0.26	0.22	0.37	0.29	0.60	0.40
**4.5**	0.07	0.07	0.09	0.09	0.12	0.12	0.17	0.16	0.24	0.21	0.34	0.27	0.57	0.38
**5**	0.06	0.07	0.09	0.09	0.11	0.12	0.15	0.15	0.22	0.20	0.32	0.26	0.54	0.38

**Table 3 children-09-01150-t003:** Centiles of SL and CV by age and sex.

	**SL (cm)**													
**Age (y)**	**C2**		**C10**		**C25**		**C50**		**C75**		**C90**		**C98**	
	**Boys**	**Girls**	**Boys**	**Girls**	**Boys**	**Girls**	**Boys**	**Girls**	**Boys**	**Girls**	**Boys**	**Girls**	**Boys**	**Girls**
**3**	36.13	40.88	40.39	43.41	43.74	45.72	47.46	48.70	51.19	52.27	54.53	56.16	58.80	62.42
**3.5**	36.95	39.85	41.17	42.45	44.48	44.78	48.16	47.74	51.85	51.20	55.16	54.84	59.38	60.44
**4**	37.90	39.54	42.04	42.26	45.29	44.66	48.90	47.60	52.52	51.06	55.77	54.54	59.91	59.68
**4.5**	39.07	40.39	43.07	43.34	46.22	45.90	49.71	49.02	53.20	52.47	56.35	55.91	60.35	60.81
**5**	40.38	41.93	44.21	45.18	47.22	47.95	50.56	51.25	53.90	54.82	56.91	58.27	60.73	63.04
	**CV of SL**													
**Age (y)**	**C2**		**C10**		**C25**		**C50**		**C75**		**C90**	**C98**		
	**Boys**	**Girls**	**Boys**	**Girls**	**Boys**	**Girls**	**Boys**	**Girls**	**Boys**	**Girls**	**Boys**	**Girls**	**Boys**	**Girls**
**3**	0.05	0.04	0.06	0.05	0.08	0.06	0.10	0.09	0.14	0.014	0.18	0.19	0.24	0.24
**3.5**	0.05	0.04	0.06	0.05	0.07	0.06	0.10	0.09	0.13	0.13	0.17	0.18	0.24	0.24
**4**	0.05	0.04	0.06	0.05	0.07	0.06	0.09	0.09	0.13	0.12	0.17	0.17	0.23	0.25
**4.5**	0.04	0.04	0.05	0.05	0.06	0.06	0.09	0.08	0.12	0.12	0.16	0.16	0.23	0.27
**5**	0.04	0.04	0.05	0.05	0.06	0.06	0.08	0.08	0.11	0.11	0.15	0.15	0.23	0.29

SL: step length; Y: years.

## Data Availability

The data that support the findings of this study are available on request from the corresponding author.

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
