# Peer review of "Reference Values of Spatial and Temporal Gait Parameters in a Contemporary Sample of Spanish Preschool Children: A Cross-Sectional Study"

_children, 2022, doi:10.3390/children9081150_

Round 1

Reviewer 1 Report

This manuscript studies the gait variability of certain parameters specifying children's gait for a cohort of Spanish preschool children. This was conducted with 383 healthy subjects with a range of ages varying from 3 to 5 years old. The output favors no significant differences by gender for the main gait parameters. In contrast, most of these parameters reduced significantly over time. The manuscript is well designed, written, and suitable for publication after minor peer review. I suggest the following:

1) By the end of the introduction, more clinical significance should be introduced; this will help the reader understand the scientific question the authors are trying to answer via this work.   

2) How you fixed the number of subjects (at least 279) is unclear. Can you elaborate more about that?  

3)  Please redo Figures 2 and 3; they are unclear; place out the legend and increase the font of the text and curves.

4) please check the typo error on the paper carefully.

Author Response

All the comments to the review are in document attached

Reviewer 2 Report

Dear authors,

The manuscript theme is very interesting. I do have some questions though:
1. The sample was composed by participants from rural and urban zones. Could this factor alter the results? Please, clarify and make a rationale in your new version.

2. You must include the IRB approval number in your text.

3. "Five kinematics parameters were analysed: cadence, SL, WR single and double support" Explain why those parameters and not other.

4. Please, explain what type of ANOVA was chosen for statistical analysis (2-way, 1-way, factorial, etc).

5. Provide details about the software R, as you did for SPSS.

6. Figure 2 and 3 are poorly explained. Please, change the approach or improve the graph's quality to enhance clarity.

7.  No info about the data availability was provided. This is very important to ensure transparency. Please, make a statement after the acknowledgements section.

Author Response

All the comments are in the document attached
